# Osmoregulatory and Antioxidants Modulation by Salicylic Acid and Methionine in Cowpea Plants under the Water Restriction

**DOI:** 10.3390/plants12061341

**Published:** 2023-03-16

**Authors:** Auta Paulina da Silva Oliveira, Yuri Lima Melo, Rayanne Silva de Alencar, Pedro Roberto Almeida Viégas, Guilherme Felix Dias, Rener Luciano de Souza Ferraz, Francisco Vanies da Silva Sá, José Dantas Neto, Ivomberg Dourado Magalhães, Hans Raj Gheyi, Claudivan Feitosa de Lacerda, Alberto Soares de Melo

**Affiliations:** 1Center of Biological and Health Sciences, Universidade Estadual da Paraíba, Campus I, Campina Grande 58429-500, PB, Brazil; 2Department of Agronomy, Federal University of Sergipe, São Cristóvão 49100-000, SE, Brazil; 3Academic Unit of Development Technology, Federal University of Campina Grande, Sumé 58540-000, PB, Brazil; 4Department of Agronomic and Forest Science, Federal Rural University of the Semi-Arid, Mossoró 59625-900, RN, Brazil; 5Center of Tecnologia and Natural Resources, Universidade Federal de Campina Grande, Campina Grande 58429-900, PB, Brazil; 6Department of Agronomy, Federal University of Alagoas, Rio Largo 57309-005, AL, Brazil; 7Department of Agronomy, Federal University of Ceará, Fortaleza 60020-903, CE, Brazil

**Keywords:** *Vigna unguiculata* (L.) Walp, antioxidant enzymes, drought tolerance

## Abstract

Global climate changes have intensified water stress in arid and semi-arid regions, reducing plant growth and yield. In this scenario, the present study aimed to evaluate the mitigating action of salicylic acid and methionine in cowpea cultivars under water restriction conditions. An experiment was conducted in a completely randomized design with treatments set up in a 2 × 5 factorial arrangement corresponding to two cowpea cultivars (BRS Novaera and BRS Pajeú) and five treatments of water replenishment, salicylic acid, and methionine. After eight days, water stress decreased the Ψw, leaf area, and fresh mass and increased the total soluble sugars and catalase activity in the two cultivars. After sixteen days, water stress increased the activity of the superoxide dismutase and ascorbate peroxidase enzymes and decreased the total soluble sugars content and catalase activity of BRS Pajeú plants. This stress response was intensified in the BRS Pajeú plants sprayed with salicylic acid and the BRS Novaera plants with salicylic acid or methionine. BRS Pajeú is more tolerant to water stress than BRS Novaera; therefore, the regulations induced by the isolated application of salicylic acid and methionine were more intense in BRS Novaera, stimulating the tolerance mechanism of this cultivar to water stress.

## 1. Introduction

Global climate changes significantly disturb the space-time distribution of rainfall, causing water deficit in agroecosystems and reducing crop growth and yield. This reality has highlighted the need for genetically improved genotypes with traits of agricultural interest and capability of adapting to different environments under adverse conditions, e.g., cowpea (*Vigna unguiculata* (L.) Walp.) [1]. Cowpea performs a critical nutritional role due to its use as human food, animal forage, and green manure. It also contributes to social and economic development in arid and semi-arid regions. In those regions, water limitation in the soil-plant-atmosphere continuum inhibits germination and plant establishment and reduces vegetative growth and grain production [2].

The tolerant ones had the highest leaf water potential when comparing tolerant and sensitive cowpea genotypes to water stress under water restriction conditions. In contrast, leaf gas exchange and chlorophyll fluorescence decreased more rapidly in the sensitive ones [3]. These differences in the physiological and biochemical responses highlight the need for studies that explore the aptitude of multiple genotypes [4]. The divergences between genotypes regarding physiological and biochemical variations and gene expression [5] suggest that the use of elicitors, i.e., low molecular weight molecules that modulate the cell metabolism, can induce responses that result in higher tolerance to water stress [6]. However, applying elicitors stimulates a cascade of biochemical reactions that modify the secondary metabolism of plant species [7]. 

Among elicitors, salicylic acid (SA) is an essential plant growth regulator that, when applied exogenously, intensifies physiological and molecular processes, including changes in gene expression, increasing protein synthesis, and activating specific enzymes [8]. SA applications from 0.1 to 1 mM increase the germination percentage and the activity of SOD, CAT, and APX in cowpea plants under drought conditions [9,10]. Araújo et al. [10] observed that applying 1 mM of SA in cowpea plants under water restriction prevented membrane damage, increased the proline content, regulated early growth, and increased the levels of chlorophyll *a*, *b*, and carotenoids. In another study, the yield of cowpea plants under water restriction conditions reached values close to 2.64–2.73 Mg ha^−1^ in the first and second crop cycles after applying 0.3 g L^−1^ of SA [8].

Methionine is an essential amino acid that participates in several physiological functions in plants. Its limitation compromises plant survival since this amino acid acts as an effective regulator in the growth and development of plants subjected to water restriction [11]. Some studies show that the exogenous foliar application of methionine is practical and positively impacts the integrity of photosynthetic pigments, the accumulation of compatible osmolytes, the removal of reactive oxygen species (ROS), and the improvement in cowpea growth and yield [12]. 

We hypothesized that foliar application of salicylic acid and methionine could modulate cowpea plants’ osmotic and antioxidant metabolism under water restriction. From this perspective, since the effects of the exogenous applications of elicitors on cowpea plants grown under stress still require explanation, the present study investigated the effects of the application of salicylic acid and methionine on the water status, growth, osmotic adjustment, oxidative stress, and leaf gas exchange indicators of two cowpea cultivars subjected to water restriction.

## 2. Results

Eight days after treatment (DAT) application, the first four PCs represented 78% of the total data variance (s^2^). PC_1_ (34% s^2^) was formed by the linear combination of the water potential (Ψw) with the leaf area (LA), shoot fresh matter (FM), the total content of soluble sugars (TSS), and catalase activity (CAT). PC_2_ (23% s^2^) is the combination between the relative leaf water content (RWC) and the content of proline (PRO), total soluble proteins (TSPs), hydrogen peroxide (H_2_O_2_), and ascorbate peroxidase activity (APX). PC_3_ (13.5% s^2^) is the combination of Ψw with superoxide dismutase activity (SOD), whereas PC_4_ (8% s^2^) is the combination of RWC with the content of total free amino acids (TFAAs). There was a significant difference (*p* < 0.01) between genotypes (G), treatments (T), and the G × T interaction in the four PCs (Table 1).

Sixteen DAT, the first four PCs explained 80% of s^2^. PC_1_ (38% s^2^) originated from the combination between RWC, LA, FM, PRO, TFAA, the CO_2_ assimilation rate (A), transpiration (E), stomatal conductance (gs), and the internal CO_2_ concentration (Ci); PC_2_ (20% s^2^) combined TSPs, APX, CAT, and SOD; PC_3_ (13% s^2^) is the combination between Ψw, TSS, and H_2_O_2_; finally, PC_4_ (9% s^2^) is the combination between TFAAs and SOD (Table 1).

After eight days, the PC1 separated treatments (left–right), and PC2 separated genotypes (lower–upper). In PC1, water stress decreased the Ψw, LA, and FM and increased the TSS and CAT activity, a process that occurred more markedly in the G2 plants (BRS Novaera) sprayed with SA + MET (T5) or when not receiving spraying (T2), and in the G1 plants (BRS Pajeú) sprayed with SA (T3) (Figure 1A,B).

In PC2, G2 showed the highest PRO accumulation and TSP content and the lowest RWC, H_2_O_2_, and APX activity compared to G1. When analyzing the effect of T within each G, the G2 plants under water stress and no spraying of elicitors (T2) and those sprayed with SA (T3) showed the highest PRO and TSP contents, similar to control plants (T1). In contrast, the plants sprayed with MET (T4) or SA + MET (T5) increased the RWC, H_2_O_2_, and APX, unlike T1 and T2. In G1, the spraying with SA, MET, and SA + MET increased the RWC, H_2_O_2_, and APX and decreased the PRO and TSPs compared to the control.

In PC3, water stress decreased the Ψw and SOD activity in G2 compared to control. However, MET spraying increased these indicators. On the other hand, the G1 plants showed increased Ψw and SOD activity under water stress, whereas spraying with SA, MET, or SA + MET maintained these indicators at levels similar to the control treatment.

In PC4, water stress decreased the RWC and TFAAs of G1 plants, with SA or MET spraying increasing these indicators in stressed plants to levels similar to the control treatment. However, when SA + MET were applied together, the RWC and TFAA levels increased but did not reach levels similar to the control treatment. G2 plants under water stress and SA spraying showed lower RWC and TFAA values, whereas MET increased these indicators to levels similar to the control treatment, and the joint application of SA + MET increased these levels to values above those of the control treatment (Figure 1C,D).

Sixteen DAT, in PC1, water stress decreased the gas exchange variables (A, E, gs, and Ci), increased the content of osmoprotectants (TFAAs and PRO), and decreased the water content (RWC) and growth (LA and FM) in the two genotypes. This process is intensified by SA or SA + MET spraying in G1 and SA in G2. Regardless of treatment, the process described above was more pronounced in G2 (Figure 2A,B and Table 2).

In PC2, water stress increased the activity of the SOD and APX enzymes and decreased the TSP content and CAT activity of G1 plants. This stress response was intensified in the G1 plants sprayed with SA and the G2 plants with SA or MET. However, foliar spraying with SA + MET significantly inhibited this response in the two genotypes and maintained enzyme activity and the TSP content at levels similar to the control plants (Figure 2A,B and Table 2).

In PC3, the two genotypes subjected to stress showed increased H_2_O_2_ contents, triggering TSS accumulation and Ψw reduction. In G1, the foliar spraying with SA or SA + MET reversed this process and made these indicators similar to control plants. In G2, the stress effect was reversed by the isolated spraying with SA or MET. However, the effect was maintained when these mitigators were applied together (Figure 2C,D and Table 2).

In PC4, spraying with SA or MET increased SOD activity and decreased the TFAA content of G1, differing from the control plants and those under stress and no spraying, which did not differ concerning these indicators. On the other hand, stress imposition in G2 increased TFAA accumulation and decreased SOD activity, a process intensified by SA or MET spraying (Figure 2C,D and Table 2).

Table 2 shows the mean values of the original variables, the scores of the principal components, and the means comparison tests for these scores.

In cowpea under water stress after 8 days of treatment (8 DAT), only ‘BRS Novaera’ had the leaf water potential decreased by 94% compared to the control. In the same period, the ‘BRS Pajeú’ under water stress showed a higher water potential (−0.52 MPa) similar to the control (−0.47 MPa) and higher than the ‘BRS Novaera’ cultivar (−0.74 MPa). Sixteen DAT, the leaf water potential of the ‘BRS Pajeú’ and ‘BRS Novaera’ cultivars decreased by 50% under water stress compared to the control. The relative leaf water content (RWC) of the two cowpea cultivars was decreased by water stress. Eight DAT, the RWC of ‘BRS Novaera’ decreased by 13.74% under water stress compared to the control. Sixteen DAT, ‘BRS Pajeú’ showed a 20% decrease in RWC under water stress compared to the control (Table 2).

Cowpea cultivars decreased leaf area (LA) under water stress compared to the control. Eight DAT, ‘BRS Pajeú’ had higher FA (98.91 cm^2^ per plant) than ‘BRS Novaera’ (79.55 cm^2^ per plant) under water stress. Sixteen DAT, the LA of ‘BRS Pajeú’ decreased by 40% and that of ‘BRS Novaera’ by 48% under water stress compared to the control. Water stress decreased fresh cowpea biomass (FM) compared to the control (Table 2). Eight DAT, ‘BRS Novaera’ (6.84 g FM) was superior to ‘BRS Pajeú’ (5.89 g FM) in water stress. Sixteen DAT, the reduction trend of FM in water stress was maintained, being 30.44% in ‘BRS Pajeú’ and 67% in ‘BRS Novaera’ compared to the control (Table 2).

In the cowpea under water stress conditions, total soluble sugars (TSS) increased by 66% and 41.2% in ‘BRS Pajeú’ and 41.4% and 27.7% in ‘BRS Novaera’ compared to the control for the 8 and 16 DAT, respectively (Table 2). In the isolated application of salicylic acid (SA, 1.5 mM), TSS decreased by 16.2% and 22.25% in ‘BRS Pajeú’ and increased by 26 and 11% in ‘BRS Novaera’ compared to water stress for 8 and 16 DAT, respectively. In the isolated application of methionine (MET, 6 mM), only ‘BRS Novaera’ increased TSS by 20.5% and 11.24% compared to water stress for 8 and 16 DAT, respectively. The proline (PRO) of ‘BRS Pajeú’ increased by 116% and 57.4% under water stress compared to the control at 8 and 16 DAT (Table 2). In the ‘BRS Novaera’, PRO increased by 774.78% and 115.28% compared to the control at 8 and 16 DAT (Table 2). The isolated application of SA decreased the PRO levels in ‘BRS Novaera’ by 46% and 38% compared to water stress at 8 and 16 DAT. The isolated application of MET (6 mM) reduced the PRO levels of ‘BRS Novaera’ by 83.14% compared to water stress at 8 DAT (Table 2).

The total free amino acids (TFAAs) of the ‘BRS Novaera’ under water stress increased 68% compared to the control at 8 DAT (Table 2). At 16 DAT, TFAAs increased by 20% in ‘BRS Pajeú’ and 11% in ‘BRS Novaera’ under water stress compared to the control. The isolated application of SA in ‘BRS Pajeú’ increased the TFAAs by 91% and 22% compared to water stress at 8 and 16 DAT. However, in ‘BRS Novaera’ with SA application, the TFAAs decreased by 46% compared to water stress at 8 DAT, and at 16 DAT, the TFAAs increased by 12% compared to water stress. The isolated application of MET (6 mM) increased the TFAA content by 50% in ‘BRS Pajeú’ at 8 DAT and 14% in ‘BRS Novaera’ at 16 DAT compared to water stress. At 8 DAT, the joint application of SA and MET increased the TFAA content in ‘BRS Novaera’ by 27% compared to water stress (Table 2).

In the cowpea under water stress conditions, the total soluble protein (TSP) content decreased by 32.26% in ‘BRS Pajeú’ and increased by 5.99% in ‘BRS Novaera’ at 8 DAT compared to the control. At 16 DAT, the TSPs of plants under water stress decreased by 34 and 36% for ‘BRS Pajeú’ and ‘BRS Novaera’ compared to the control. The isolated application of SA (1.5 mM) compared to water stress increased TSPs by 43 and 21% for ‘BRS Pajeú’ and ‘BRS Novaera’ at 8 DAT. At 16 DAT, the isolated application of SA (1.5 mM) decreased TSPs by 43.66% and 21% for ‘BRS Pajeú’ and ‘BRS Novaera’ compared to water stress (Table 2).

In plants under water stress at 8 DAT, H_2_O_2_ was increased by 65% in ‘BRS Pajeú’ compared to the control (Table 2). However, at 16 DAT, H_2_O_2_ increased by 46% in ‘BRS Pajeú’ and 51% in ‘BRS Novaera’ compared to the control. At 8 DAT, the isolated application of SA increased H_2_O_2_ by 15% in ‘BRS Pajeú’ and 38% in ‘BRS Novaera’ compared to water stress. However, in the isolated application of SA at 16 DAT, the ‘BRS Pajeú’ and ‘BRS Novaera’ cultivars decreased the H_2_O_2_ content by 26 and 28% compared to water stress. In the isolated application of MET compared to water stress at 8 DAT, H_2_O_2_ increased by 23% in ‘BRS Pajéu’ and 42% in ‘BRS Novaera’ (42%). At 16 DAT with MET application, H_2_O_2_ increased by 30% only in ‘BRS Pajeú’ compared to water stress (Table 2).

The ascorbate peroxidase activity (APX) of plants under water stress at 8 DAT decreased by 24.5% in ‘BRS Pajeú’ and increased by 93% in ‘BRS Novaera’ compared to the control. At 16 DAT, APX activity decreased by 18% and 37% for ‘BRS Pajeú’ and ‘BRS Novaera’ compared to the control. The foliar application of SA (1.5 mM) at 8 DAT increased APX activity by 11% and 34% for ‘BRS Pajeú’ and ‘BRS Novaera’ compared to water stress. At 16 DAT, SA increased APX activity by 78% in ‘BRS Pajéu’ and 670% in ‘BRS Novaera’ compared to water stress. Foliar application of MET (6 mM) at 16 DAT increased APX activity by 570% only for ‘BRS Novaera’, compared to water stress. The joint application of SA and MET increased APX activity by 222% for ‘BRS Novaera’ at 16 DAT compared to water stress (Table 2).

The catalase (CAT) activity of plants under water stress at 8 DAT increased by 20.63% for ‘BRS Pajeú’ and 66.15% for ‘BRS Novaera’ compared to the control. However, at 16 DAT, the CAT activity of plants under water stress decreased by 26% in ‘BRS Novaera’ compared to the control. The isolated application of SA (1.5 mM) at 8 DAT increased CAT activity by 53% for ‘BRS Pajeú’ and 15% for ‘BRS Novaera’, compared to the water stress. However, the isolated application of SA (1.5 mM) at 16 DAT decreased CAT activity by 41% for ‘BRS Pajeú’ and 40% for ‘BRS Novaera’ compared to the water stress. At 8 DAT, MET foliar application (6 mM) increased CAT activity by 70% in ‘BRS Pajeú’ and 39% in ‘BRS Novaera’ compared to water stress. However, at 16 DAT, MET foliar application increased 16.35% CAT activity in ‘BRS Pajeú’ compared to the water stress. In the joint application of SA and MET compared to water stress, CAT activity increased by 46% for ‘BRS Pajeú’ and 23% for ‘BRS Novaera’ at 8 DAT. However, at 16 DAT, CAT activity increased by 19% only for ‘BRS Novaera’ compared to water stress (Table 2).

In plants under water stress at 8 DAT, superoxide dismutase (SOD) activity increased by 136% for ‘BRS Pajeú’ and 118% for ‘BRS Novaera’ compared to the control. At 16 DAT, the SOD activity of plants under water stress decreased by 45% for ‘BRS Novaera’ compared to the control (Table 2). At 8 DAT, the SA application compared to water stress decreased SOD activity in ‘BRS Pajeú’ by 73%. At 16 DAT, the SA application compared to water stress decreased increased SOD activity by 51% for ‘BRS Pajeú’ and 57% for ‘BRS Novaera’. Foliar application of MET (6 mM) at 8 DAT compared to water stress increased SOD activity by 220% in ‘BRS Novaera’ and decreased SOD activity by 74% in ‘BRS Pajeú’ (Table 2).

Leaf gas exchange was evaluated only at 16 DAT, and all leaf gas exchange variables of ‘BRS Pajeú’ decreased under water stress, with a decrease of 47% in photosynthesis, 47% in transpiration, 59% in stomatal conductance, and 37% in carbon internal compared to the control. In ‘BRS Novaera’ under water stress, internal carbon increased by 66% compared to the control, while photosynthesis, transpiration, and stomatal conductance decreased by 47%, 29%, and 42% compared to the control. The foliar application of SA (1.5 mM) decreased photosynthesis by 7.35% and 24%, transpiration by 30% and 41%, and stomatal conductance by 16% and 34.17% for ‘BRS Pajeú’ and ‘BRS Novaera’ compared to water stress, respectively. However, only the internal carbon of ‘BRS Novaera’ decreased by 33% compared to water stress. The foliar application of MET (6 mM) increased photosynthesis by 5.53%, transpiration by 6.36%, and stomatal conductance of ‘BRS Pajéu’ by 6.03% compared to water stress. However, in ‘BRS Novaera’, the application of MET decreased photosynthesis by 17.68%, transpiration by 28.57%, and stomatal conductance by 24.05% compared to water stress. In the joint application of SA and MET, only the internal carbon of ‘BRS Novaera’ increased by 28.24% compared to water stress (Table 2).

## 3. Discussion

Some cowpea cultivars (e.g., BRS Novaera) show reductions in the leaf water potential under water restriction conditions, damaging different growth indicators, including leaf area and fresh mass [13]. One of the effects most related to the reduction in growth indicators caused by water restriction is the loss of cell turgor, which restricts cell division and elongation and causes physiological changes that include disturbances in photosynthesis [14]. 

The probable damage caused to the photosynthetic apparatus might have favored an increase in the levels of reactive oxygen species in the cultivar BRS Novaera as soon as the first eight days of stress, justifying the increase in the activity of antioxidant enzymes such as CAT. The increase in the activity of this enzyme is due to the drought tolerance mechanism attributed to some cowpea cultivars [6]. In the cultivar BRS Novaera, stress imposition for eight weeks increased the TSS content in cowpea leaves, which is considered an expected reaction and could be related to starch degradation, favoring the action of the osmolyte as an osmotic adjuster or metabolic signaling molecule frequently involved with drought tolerance [15,16]. 

In addition to TSS, other compatible osmolytes, e.g., PRO and TSPs, can contribute to the osmotic adjustment process of different species under stress conditions, including cowpea [17,18]. In addition to the cultivar BRS Novaera, an increase in PRO levels after SA application (T3) was already observed in other cowpea cultivars by Andrade et al. [19]. SA is related to several regulatory functions of plant metabolism and activates defense mechanisms against water deficit, including osmotic adjustment [20]. Compatible solutes are typically hydrophilic and can replace water molecules on the surface of proteins and membranes, which increases osmotic pressure and the potential water gradient between soil and roots at the first moment, thus enabling a continuous influx of water by osmosis throughout the plant [17]. 

In the cowpea cultivars BRS Pajeú (G1) and BRS Novaera (G2), the application of plant elicitors eight days after the beginning of the water deficit increased the RWC, H_2_O_2_, and APX activity. The action mechanism of SA suggests that this acid is also responsible for increasing the concentrations of ROS, such as H_2_O_2_, during the first days of stress through a signaling process that leads to the activation of the cellular detoxification mechanism, thus promoting stress tolerance [21]. Applying both SA and MET can intensify the activity of antioxidant enzymes under water deficit conditions, maintaining membrane stability and increasing the plant water status [11,12]. Specifically, in the cultivar BRS Pajeú (G1), the application of elicitors might have directly contributed to water status maintenance processes unrelated to the accumulation of compatible osmolytes, e.g., PRO and TSPs, even during the first evaluation period. 

Since the enzyme activity observed results from both synthesis and degradation, the net SOD activity in the cultivar BRS Novaera (G2) reduced after 16 days under water deficit. According to Liang et al. [22], a decrease in the SOD synthesis or an increase in the SOD proteolysis occurs in plants under water deficit due to disturbances in the photosynthetic mechanism. Furthermore, H_2_O_2_ accumulation under drought conditions can also decrease SOD activity. After eight days, MET application in the cultivar BRS Novaera (G2) and the remaining elicitors in BRS Pajeú (G1) increased the SOD levels and contributed to the recovery of the plant water status. Both MET and SA effectively removed the superoxide ion, primarily through increased SOD activity [9,11]. The increase in this enzyme’s activity contributes to maintaining membrane integrity by reducing lipid peroxidation [22]. 

In the present study, the water deficit decreased the TFAA levels in both cowpea cultivars after eight days. According to Goufo et al. [17], cowpea can regulate its nitrogen metabolism, converting amino acids into proteins and vice versa, depending on the current needs of the plant. Therefore, the results of the present study suggest that reduced TFAA levels could be related to increased TSP concentrations, especially in the cultivar BRS Novaera (G2). The application of elicitors increased the TFAA levels of the cultivars BRS Pajeú (G1) and BRS Novaera (G2) under water restriction conditions in the first eight days. Due to the apparent plasticity of cowpea plants in controlling their N metabolism, increases in the TFAA levels are relevant not only for protein biosynthesis but also for influencing different physiological processes, e.g., growth and development. Gorni et al. [23] indicate that TFAA levels contribute to intracellular pH control, metabolic energy generation, and plant water stress tolerance. 

After 16 days of stress, the plasticity of cowpea plants in regulating their morphophysiological attributes became clear. However, the photosynthetic metabolism of the species is sensitive to water deficit since the reduction in net photosynthesis occurs by stomatal closure, limiting the CO_2_ supply for RuBisCO as soon as the stress is imposed [6]. In general, cowpea cultivars under water deficit show a rapid reduction in stomatal conductance, which signals the closing of stomata, followed by low photosynthetic and transpiration rates [3]. Nevertheless, stomatal closure is one of the first responses of the species, working as an efficient adaptative mechanism to control transpiration. The increase in the osmoregulatory levels such as PRO and TFAAs, in turn, could work as an osmotic adjustment in some cowpea cultivars under water restriction [3,13,17,24], which, in the present study, also happened after 16 days of stress. The active mechanism works through the synthesis and accumulation of organic solutes in the cytoplasm, and the result is the reduction in the water potential of the plant, providing a water potential gradient favorable to water uptake and the maintenance of cell turgor [17,25].

The prolonged water stress of 16 days (16 DAT) might have intensified the disturbances in photosynthetic processes and affected the growth indicators of cowpea, e.g., dry matter production and leaf area expansion [19,26] and pod weight and yield [27]. Although SA is frequently associated with beneficial effects on photosynthetic processes even under water deficit conditions, the second application of elicitors in both cultivars of the present study seems to have negatively affected the gas exchange indicators. The second application of 0.21 g L^−1^ SA in the 8-day interval may have generated momentary photosynthetic disturbances during the evaluated period. Kumar et al. [28] reported the harmful effects on the photosynthetic activity and the inhibition of the nitrate uptake system in *Trifloium alexandrinum* (L.) after the application of a high level of salicylic acid (100 μg mL^−1^).

The increased ROS levels observed after 16 days of stress, especially in the cultivar BRS Pajeú (G1), could result in lipid peroxidation, protein oxidation, inhibited enzyme activity, oxidative damage to RNA and DNA, and cell death [11,29]. In the present study, the modulation in the biochemical mechanism of cowpea plants under water restriction becomes more evident over time. After 16 days under stress, the increase in the levels of antioxidant enzymes in both cultivars suggests the intensification of the antioxidant metabolism mediated by SOD, which converts O^2−^ into H_2_O_2_ so that this oxidizing agent can be more easily converted into H_2_O, preferably by the enzyme APX at the second moment, reducing the oxidative damage to plant cells [19]. In the referred period, the intensification of SOD and APX activity after SA application in both cultivars and MET application in the cultivar BRS Novaera (G2), under water restriction, was also observed by Andrade et al. [19] and Merwad et al. [12] in other cowpea cultivars. 

After 16 days under water deficit, stress continued to restrict the water supply to leaf tissues, causing oxidative damage to cowpea plants, represented by the increased H_2_O_2_ levels. As a result, high H_2_O_2_ levels can interrupt the photosynthetic machinery [30]. However, changes to the gene expression and protein levels are also triggered during the stress period [31], inducing increases in cowpea [32]. The application of elicitors in both cultivars, especially SA, enabled plants to resume the water supply and balance the oxidative metabolism by increasing the leaf water potential and reducing the H_2_O_2_ levels [11]. Salicylic acid protects the cell membranes and their carrier proteins, which maintain their structure and function against the toxic and disruptive effects of reactive oxygen species released during stress [33]. At a higher intensity in the cultivar BRS Novaera (G2), MET application after 16 DAT effectively reduced the H_2_O_2_ contents in stressed plants, which may have contributed to reducing the lipid peroxidation levels and, consequently, maintaining the damage to cell membranes [11,12]. 

The divergences observed between the cowpea cultivars BRS Pajeú (G1) and BRS Novaera (G2) after 16 DAT for the SOD and TFAA contents under stress conditions in the absence and presence of elicitors highlight the evidence that the responses of plants of the same species to stress conditions depend on each genotype [4,34]. In the cultivar BRS Pajeú (G1), water restriction may have reduced nitrogen assimilation even after the second application of elicitors [12] since this nutrient is an indispensable intermediate in the nitrogen metabolism and the biosynthesis of amino acids. After SA application, the intensified reduction in TFAA levels indicates the participation of the elicitor in inducing the expression of 11 new cowpea proteins in plants subjected to water stress, which is related to the improvement in growth and production [8]. On the other hand, in the cultivar BRS Novaera (G2) under water restriction for 16 days, the increase in the TFAA levels could be related to the osmotic adjustment process, in which the osmoprotectant mediated by these molecules reduced the damage to cell membranes and, consequently, the levels of ROS, possibly justifying the reduction in antioxidant activity. SA application intensified the synthesis of TFAAs and highlighted the importance of these molecules in different anti-stress metabolic pathways since they act as signaling molecules and precursors for synthesizing plant hormones or secondary metabolites of the defense mechanism [23].

## 4. Materials and Methods

### 4.1. Location of the Research Area and Experimental Design

The experiment was conducted from October to December 2019 at the Forest Garden (Horto Florestal) area (an extension of the Integrated Research Complex of Três Marias, Campus I) of the State University of Paraíba, located in Campina Grande—PB (07°13′50″ S, 35°52′52″ W, at an elevation of 551 m). The climate is classified as *BSh*, according to Köppen and Geiger, with a mean temperature of 23.3 °C and a mean annual rainfall of 503 mm [35].

We subjected two cowpea cultivars (BRS Novaera and BRS Pajeú) to five treatments, described as follows: control (T_1_: 100% of soil field capacity); stress (T_2_: 50% of soil field capacity); T_3_: stress + salicylic acid application − SA (0.21 g L^−1^); T_4_: stress + methionine application − MET (0.89 g L^−1^); T_5_: stress + SA (0.21 g L^−1^) + MET (0.89 g L^−1^). The SA and MET concentrations were established based on previous studies conducted by Dutra et al. [9] and Merwad et al. [12]. The experiment was set up in a completely randomized design with five replications (n = 5), totaling fifty experimental units composed of two plants per pot.

### 4.2. Conduction of the Experiment

Pots with a capacity of 3.6 dm^3^ were filled with clayey-sandy soil whose physical and fertility characteristics were as follows: sand: 659 g kg^−1^; silt: 101 g kg^−1^; clay: 240 g kg^−1^; apparent density: 1.38 kg dm^−3^; particle density: 2.63 kg dm^−3^; total porosity: 0.48 m^3^ m^−3^; calcium: 2.38 cmol_c_ dm^−3^; magnesium: 1.66 cmol_c_ dm^−3^; sodium: 0.23 cmol_c_ dm^−3^; potassium: 0.14 cmol_c_ dm^−3^; hydrogen + aluminum: 5.69 cmol_c_ dm^−3^; and organic matter: 20.38 g kg^−1^; pH: 4.8.

The soil was corrected to increase the base saturation percentage to 70% using 4.8 mg CaCO_3_ (pure for analysis) per pot. After correction, the soil was moist for 30 days before sowing. We applied 20 kg ha^−1^ of P_2_O_5_, 30 kg ha^−1^ of N, and 35 kg ha^−1^ of K_2_O [36], corresponding to 36 mg of P_2_O_5_, 54 mg of N, and 63 mg of K_2_O per pot. The fertilizers used were urea (45% N), monoammonium phosphate (12% N and 65% P_2_O_5_), and potassium chloride (60% K_2_O). Monoammonium phosphate was applied before sowing. Urea and potassium chloride were applied in equal portions at 30°, 47°, and 52° days after sowing. The seeds used in the experiment were processed to remove those with physical damage and malformations. Subsequently, the seeds were exposed to the preventive fungicide Captan^®^ (Captana 800 g kg^−1^ a.i) at 0.11 g per 100 g^−1^ of seeds and were left to rest for 24 h. Soil saturation was performed before sowing for 48 h, followed by excess water drainage to maintain the soil at field capacity on the sowing [37]. The pots were arranged in five rows with ten pots each and spaced 0.8 m between tows and 0.6 m between pots. 

Sowing was performed by equidistantly distributing five seeds per pot at a mean depth of two centimeters. The plants were thinned to two seedlings per pot 17 days after sowing. Water replenishment was performed daily and manually from sowing to the beginning of treatments. This procedure was performed according to the water volume lost by evapotranspiration in each pot, with irrigation using 100 and 50% of the water volume lost by evapotranspiration in the control and other treatments (stress, stress + SA, stress + MET, and stress + SA + MET), respectively.

Water replenishment was performed daily by weighing the pots and calculating the water volume to be replenished based on the difference between the maximum storage– AM (water capacity available in the pot) and the current storage—AT, according to Casaroli and van Lier [38], using the following equation:I = AM − AT(1)
where:I—necessary irrigation, L pot^−1^;AM—maximum storage, L pot^−1^;AT—current storage, L pot^−1^.

When the plants reached the developmental stage V8 (55 days after sowing—DAS), the application of treatments began with water restriction and the first application of elicitor substances at a level of 20 mL per plant, totaling 40 mL per pot for each solution (SA, MET, and SA + MET). The elicitors were applied via spraying until the runoff point of the solution in the abaxial and adaxial regions of the leaves. 

After eight days of treatment (8 DAT), one plant from each pot was collected for destructive analysis using the following parameters: water status (water potential—Ψw; relative water content—RWC), growth (leaf area—LA and total fresh mass—FM), osmoregulators (total soluble sugars—TSS, proline—PRO, total free amino acids—TFAAs, total soluble proteins—TSPs), and the antioxidant mechanism (hydrogen peroxide—H_2_O_2_, superoxide dismutase—SOD, ascorbate peroxidase—APX, and catalase—CAT). The treatments were reapplied in the plants that remained in the pots 8 DAT. After eight days of the reapplication, the gas exchange variables were measured at 16 DAT, followed by sample collection to measure these variables.

### 4.3. Growth Analysis and Plant Water Status

The Ψw measurements in the plant petioles were performed with a Scholander 3005F01 pressure chamber (Soil Moisture Corp., Santa Barbara, CA, USA) from 3:00 to 5:00 a.m., with values expressed as MPa [39]. For the RWC analysis, three fresh leaf disks were removed with a copper cutter and weighed (DFM), immersed in 10 mL of distilled water for 24 h, and weighed again to obtain the turgid mass (DTM). Then, the material was dried in a forced air oven at 80 °C for 24 h to obtain its dry mass (DDM). The RWC was calculated using the equation proposed by Smart and Bingham [40]:
(2)$$RWC\left(\%\right)=\frac{\left(DFM-DDM\right)}{\left(DTM-DDM\right)}\times 100$$where: RWC (%) = relative water content. DFM = fresh disk mass. DTM = turgid disk mass. DDM = dry disk mass.

The leaf area (LA) was determined using a planimeter (LI-3100, USA) and expressed as cm^2^. The fresh leaf mass (FM) was obtained by weighing the plant material in an analytical balance.

### 4.4. Analysis of Compatible Osmotic Concentrations

TSS quantification in leaf blades was performed by the phenol-sulfuric acid method described by Dubois et al. [41], with readings in a spectrophotometer at 490 nm of absorbance and expressed as mg TSS g^−1^ of FM. PRO quantification in leaf blades was performed by the colorimetric method proposed by Bates et al. [42], with readings at 520 nm. The concentrations were based on the L-proline standard curve and were expressed as µmol of PRO g^−1^ of FM.

The TFAA concentration in leaf blades was determined according to the method described by Peoples et al. [43], with readings at 570 nm quantified according to the glutamine standard curve and expressed as µmol TFAA g^−1^ FM. The TSP concentration was determined according to Bradford [44], with readings at 595 nm and data described as mg TSP g^−1^ FM using the albumin standard curve as a reference.

#### Analysis of Antioxidant Mechanism Components

H_2_O_2_ quantification in leaf blades was performed following the method of Velikova et al. [45], with readings at 390 nm and concentrations expressed as µmol H_2_O_2_ g^−1^ FM. SOD activity in leaf blades was determined based on the photoreduction inhibition capacity of nitro blue tetrazolium chloride (NBT) by the enzyme of the plant extract, following the method of Beauchamp and Fridovich [46] and expressed as U min^−1^ mg^−1^ of protein, with readings at 560 nm of absorbance. CAT activity in leaf blades was quantified according to Kar and Mishra [47], with readings at 240 nm expressed as µmol of H_2_O_2_ min^−1^ mg^−1^ of protein. 

APX activity in leaf blades was determined by the method proposed by Nakano and Asada [48], calculated based on ascorbate consumption by monitoring the decrease in absorbance in 10 readings at 290 nm in a quartz cuvette, expressed as nmol of ascorbate min^−1^ mg^−1^ of protein. 

### 4.5. Leaf Gas Exchange

The gas exchange variables were measured in the middle region of the plant, represented by net photosynthesis—A (µmol CO_2_ m^−1^ s^−1^), transpiration—E (mmol H_2_O m^−1^ s^−1^), stomatal conductance—gs (mmol H_2_O m^−1^ s^−1^), and internal CO_2_ concentration—Ci (ppm) using an infrared gas analyzer—IRGA (Infra-red Gas Analyzer)—GFS 3000 FL. All measurements were performed in the morning, between 8:00 and 11:00 a.m., in a leaf area of 8 cm^2^ using an artificial radiation source with an intensity of 1200 μmol m^−2^ s^−1^ under control cuvette conditions at a temperature of 26.4 °C (± 1), relative humidity of 60% (±1), and CO_2_ concentration of 400 µmol mo1^−1^.

### 4.6. Statistical Analysis

The data on the original variables were standardized to obtain the zero mean and unit variance ($\overset{-}{x}$ = 0 and s^2^ = 1) and evaluated by principal component analysis (PCA) and multivariate analysis (MANOVA) by Hotelling’s test (*p* ≤ 0.05). The means of the scores of each principal component (PC) for the cultivation conditions were compared by Tukey’s test (*p* ≤ 0.05), and the means of the cultivars were compared by Student’s *t*-test (*p* ≤ 0.05).

## 5. Conclusions

In the present study, exogenous applications of salicylic acid and methionine in cowpea plants under water restriction modulated the osmoregulation metabolism of free amino acids, proline, soluble proteins, and free carbohydrates, in addition to the antioxidant activity of the SOD, APX, and CAT enzymes, improving the water status after eight days of application of treatments. Furthermore, after 16 days, the cowpea cultivars under stress showed reduced water loss by transpiration, which may have regulated the photosynthetic processes. 

The regulations in the cultivar BRS Novaera induced by salicylic acid and methionine were more expressive and beneficial to intensifying the tolerance mechanism of the species to water restriction. Therefore, both elicitors act as modulators of the species’ metabolism and effectively mitigate the effects of water stress. Further research is required to identify the influence of elicitors in the reproductive phase and their consequent representativeness in the production indicators of the species, including under field conditions.

## Figures and Tables

**Figure 1 plants-12-01341-f001:**
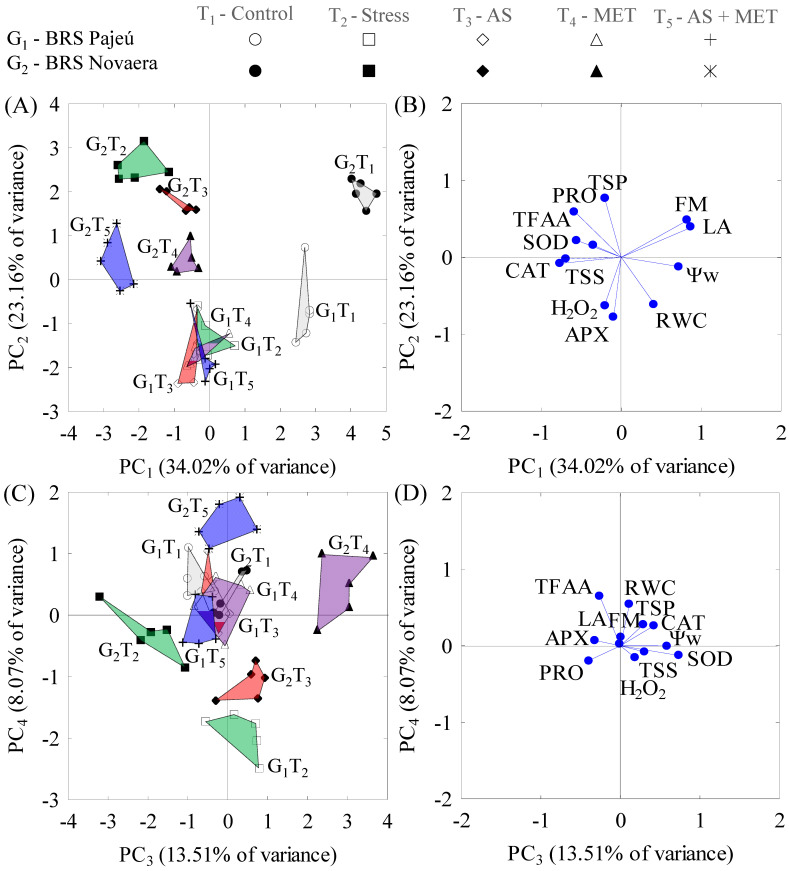
Two-dimensional projection of the interactions between genotypes and treatments (**A**,**C**) and between the correlation coefficients of the variables (**B**,**D**) with the first four principal components (PCs 1, 2, 3, and 4) eight days after the application of treatments.

**Figure 2 plants-12-01341-f002:**
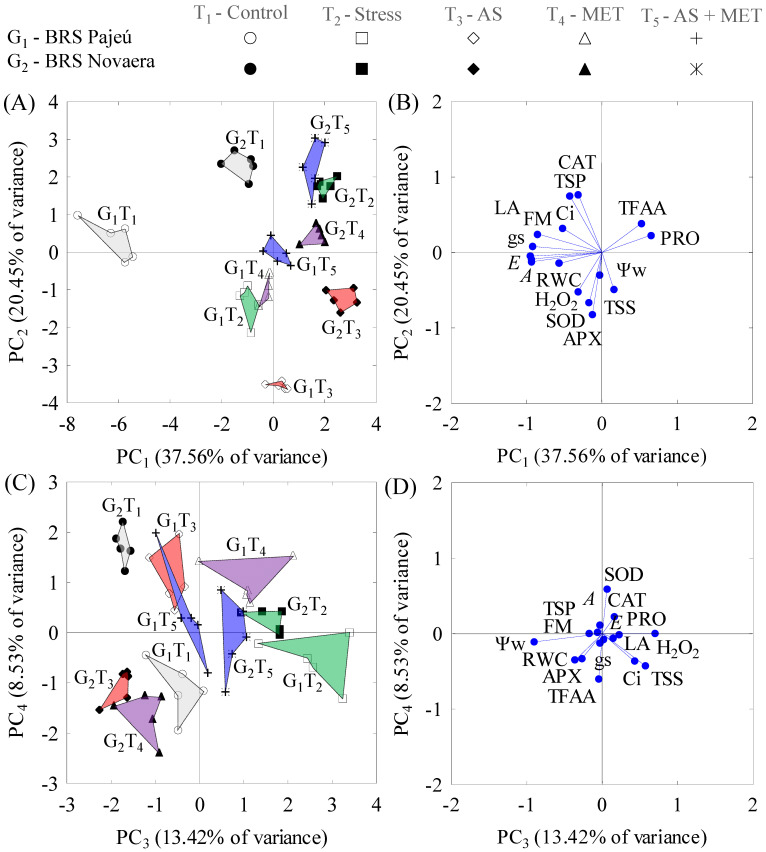
Two-dimensional projection of the interactions between genotypes and treatments (**A**,**C**) and between the correlation coefficients of the variables (**B**,**D**) and the first four principal components (PCs 1, 2, 3, and 4) 16 days after application of treatments.

**Table 1 plants-12-01341-t001:** Correlation between original variables and principal components, eigenvalues, explained and cumulative variances, and the probability of significance of the test of hypothesis after eight and 16 days of the application of treatments.

Variables	Principal Components							
	8 Days after Treatments				16 Days after Treatments			
	PC_1_	PC_2_	PC_3_	PC_4_	PC_1_	PC_2_	PC_3_	PC_4_
Ψw—Leaf water potential	0.73 *	−0.12	0.58 *	−0.00	−0.03	−0.31	−0.90 *	−0.11
RWC—Relative leaf water content	0.42	−0.61 *	0.12	0.55 *	−0.58 *	−0.15	−0.36	−0.35
LA—Leaf surface area	0.88 *	0.39	−0.01	0.03	−0.92 *	0.08	−0.03	−0.13
FM—Shoot fresh matter	0.83 *	0.48	0.01	0.12	−0.85 *	0.25	−0.06	0.02
TSS—Total soluble sugars	−0.77 *	−0.08	0.30	−0.08	0.16	−0.50	0.58 *	−0.43
PRO—Proline content	−0.58	0.59 *	−0.40	−0.2	0.65 *	0.23	0.22	−0.01
TFAAs—Total free amino acids	−0.55	0.22	−0.26	0.65 *	0.53 *	0.39	−0.05	−0.61 *
TSPs—Total soluble proteins	−0.21	0.76 *	0.29	0.28	−0.42	0.74 *	−0.17	0.00
H_2_O_2_—Hydrogen peroxide	−0.21	−0.63 *	0.18	−0.15	−0.32	−0.52	0.70 *	−0.00
APX—Ascorbate peroxidase activity	−0.09	−0.78 *	−0.32	0.07	−0.13	−0.82 *	−0.27	−0.33
CAT—Catalase activity	−0.69 *	−0.02	0.42	0.26	−0.31	0.76 *	0.17	0.23
SOD –Superoxide dismutase activity	−0.35	0.16	0.74 *	−0.13	−0.17	−0.66 *	0.06	0.58 *
A—Net CO_2_ assimilation rate	ne	ne	ne	ne	−0.94 *	−0.12	−0.04	0.11
E—Water vapor transpiration rate	ne	ne	ne	ne	−0.94 *	−0.09	0.15	−0.06
gs—Stomatal conductance to water vapor	ne	ne	ne	ne	−0.96 *	−0.05	0.02	−0.08
Ci—Internal carbon dioxide concentration	ne	ne	ne	ne	−0.52 *	0.32	0.42	−0.37
λ—Eigenvalues	4.08	2.78	1.62	0.97	6.01	3.27	2.15	1.37
S^2^ (%)—Explained variance	34.02	23.16	13.51	8.07	37.56	20.45	13.42	8.53
S^2^ (%)—Cumulative variance	34.02	57.18	70.69	78.77	37.56	58.00	71.42	79.96
MANOVA—Multivariate analysis of variance	*p*-value							
Hotelling’s T-squared test for genotypes—G	<0.01	<0.01	<0.01	<0.01	<0.01	<0.01	<0.01	<0.01
Hotelling’s T-squared test for treatments—T	<0.01	<0.01	<0.01	<0.01	<0.01	<0.01	<0.01	<0.01
Hotelling’s T-squared test for the G × T interaction	<0.01	<0.01	<0.01	<0.01	<0.01	<0.01	<0.01	<0.01

*: Variable with Pearson’s correlation coefficient (r ≥ 0.55) considered for PC; ne: not evaluated eight days after the application of treatments; PC: principal component.

**Table 2 plants-12-01341-t002:** Means of the original variables and scores of the principal components eight and sixteen days after application of treatments.

Var	*p*-Value	Means of the Combinations between Genotypes and Treatments after 8 Days										
		G1—BRS Pajeú						G2—BRS Novaera				
		T1	T2	T3	T4	T5		T1	T2	T3	T4	T5
Ψw	<0.01	−0.47Ba	−0.52Aa	−0.53Aa	−0.57Ba	−0.56Aa		−0.38Aa	−0.74Bc	−0.56Ab	−0.43Aa	−0.65Bbc
RWC	<0.01	93.43Aa	82.84Ac	86.98Abc	88.37Aabc	89.63Aab		86.53Ba	74.97Bb	74.64Bb	87.75Aa	86.00Aa
LA	<0.01	156.80Ba	98.91Ab	77.92Bc	75.04Ac	84.62Abc		260.29Aa	79.55Bc	116.93Ab	85.08Ac	73.44Bc
FM	<0.01	11.74Ba	5.89Bb	5.13Bbc	5.55Bbc	4.66Ac		15.39Aa	6.84Ac	8.22Ab	7.03Ac	5.34Ad
TSS	<0.01	12.05Ad	20.01Aa	16.78Bbc	14.59Bc	19.05Bab		10.94Ad	15.91Bc	20.01Ab	19.17Ab	24.21Aa
PRO	<0.01	8.61Ab	18.64Ba	10.17Bab	6.76Ab	12.03Bab		6.95Ac	60.71Aa	32.82Ab	10.23Ac	28.89Ab
TFAAs	<0.01	0.48Ab	0.35Bc	0.66Aa	0.52Ab	0.42Bbc		0.45Acd	0.77Ab	0.40Bd	0.54Ac	0.94Aa
TSPs	<0.03	5.92Bab	4.01Bb	5.78Bab	6.73Ba	4.79Bb		7.85Ab	8.32Aab	10.03Aa	8.95Aab	8.50Aab
H_2_O_2_	<0.01	8.12Ac	13.37Ab	15.39Aa	16.41Aa	7.59Bc		6.98Bb	6.89Bb	9.50Ba	9.77Ba	9.35Aa
APX	<0.06	65.70Aa	49.61Aa	54.99Aa	49.07Aa	70.82Aa		13.46Bb	26.00Bab	34.86Bab	26.00Bab	47.71Ba
CAT	<0.50	1.26Ac	1.53Bbc	2.32Aa	2.59Aa	2.25Aab		1.31Ac	2.16Ab	2.49Aab	3.01Aa	2.66Aab
SOD	<0.01	7.69Ab	18.16Aa	4.72Bb	4.67Bb	5.58Bb		4.18Ac	9.10Bbc	13.16Ab	29.07Aa	14.54Ab
Means comparison test for the scores of the principal components												
PC1	<0.01	1.35Ba	−0.01Ab	−0.23Ab	−0.13Ab	−0.05Ab		2.14Aa	−1.02Bc	−0.41Ab	−0.33Ab	−1.31Bc
PC2	<0.01	−0.41Ba	−0.79Bab	−1.04Bb	−1.01Bb	−1.04Bb		1.19Aa	1.53Aa	1.06Aa	0.26Ab	0.25Ab
PC3	<0.01	−0.61Bb	0.30Aa	−0.26Bab	−0.04Bab	−0.52Ab		0.02Ab	−1.54Bc	0.44Ab	2.26Aa	−0.05Ab
PC4	<0.01	0.61Aa	−1.97Bc	0.34Aab	0.24Aab	−0.14Bb		0.33Abc	−0.31Ac	−1.12Bd	0.48Ab	1.53Aa
		Means of the combinations between genotypes and treatments after 16 days										
Ψw	<0.01	−0.28Aa	−0.42Ac	−0.25Ba	−0.36Bb	−0.26Aa		−0.28Ab	−0.42Ac	−0.20Aa	−0.26Ab	−0.37Bc
RWC	<0.01	81.95Aa	73.67Ab	71.18Bb	74.28Bb	71.79Ab		74.40Ba	69.90Bb	74.49Aa	77.86Aa	64.83Bc
LA	<0.01	267.48Aa	149.73Ab	129.45Ab	92.17Bc	125.22Ab		144.89Ba	75.38Bc	58.85Bc	111.86Ab	130.74Aab
FM	<0.01	15.24Aa	10.60Ab	7.95Ac	6.30Ac	7.78Ac		12.52Ba	7.20Bbc	5.73Bc	7.07Abc	8.33Ab
TSS	<0.01	14.30Ac	20.20Aa	15.70Abc	16.64Ab	16.16Abc		11.34Bb	14.49Ba	16.02Aa	16.12Aa	15.86Aa
PRO	<0.01	9.04Bc	14.23Bbc	14.61Babc	17.46Aab	20.40Aa		14.14Ac	30.43Aa	18.85Abc	20.70Ab	16.90Abc
TFAAs	<0.01	0.77Aab	0.91Aa	0.71Bb	0.74Bab	0.81Bab		0.87Ab	0.96Aab	1.07Aa	1.08Aa	1.05Aab
TSPs	<0.27	9.69Aa	6.40Ab	3.61Bc	5.36Bb	8.62Aa		10.27Aa	6.58Abc	5.21Ac	7.10Ab	9.16Aa
H_2_O_2_	<0.01	8.12Ab	11.84Aa	8.54Ab	8.27Ab	7.61Ab		5.18Bb	7.82Ba	5.81Bb	5.81Bb	5.33Bb
APX	<0.01	90.00Ab	73.81Abc	131.58Aa	59.14Bcd	54.25Ad		20.32Bc	12.86Bc	98.98Ba	86.13Aa	41.46Bb
CAT	<0.01	3.03Aa	2.69Aa	1.59Ab	3.14Aa	2.47Ba		3.88Aa	2.87Ab	1.71Ac	2.72Ab	3.43Aab
SOD	<0.01	12.28Ac	16.74Abc	25.21Aa	20.10Ab	13.27Ac		14.34Aa	7.94Bbc	12.49Bab	6.91Bc	12.10Aab
A	<0.01	23.28Aa	12.17Abc	11.48Ac	13.06Ab	13.38Ab		14.05Ba	7.33Bb	5.65Bc	6.08Bbc	3.86Bd
E	<0.01	3.11Aa	1.65Ab	1.16Ac	1.76Ab	1.20Ac		1.39Ba	0.98Bb	0.58Bc	0.71Bbc	0.53Bc
gs	<0.01	157.85Aa	65.50Ab	55.30Ab	69.45Ab	57.05Ab		68.25Ba	39.50Bb	26.00Bb	30.00Bb	22.00Bb
Ci	<0.01	132.05Aa	83.30Ab	56.75Ab	83.70Ab	70.05Bb		52.45Bc	87.10Aab	58.59Ac	69.80Abc	111.70Aa
Means comparison test for the scores of the principal components												
PC1	<0.01	−2.51Bc	−0.38Bb	0.11Ba	−0.09Bab	0.08Ba		−0.49Ac	0.84Aab	1.11Aa	0.69Ab	0.65Ab
PC2	<0.01	0.18Ba	−0.74Bb	−1.97Bc	−0.52Bb	−0.02Ba		1.28Aa	0.97Aa	−0.69Ac	0.25Ab	1.26Aa
PC3	<0.01	−0.33Ac	1.78Aa	−0.43Ac	0.74Ab	−0.19Bc		−1.17Bb	1.08Ba	−1.20Bb	−0.81Bb	0.54Aa
PC4	<0.01	−0.97Bc	−0.48Bbc	0.95Aa	0.88Aa	0.32Aab		1.47Aa	0.21Ab	−0.91Bcd	−1.39Bd	−0.08Abc

Var: variables; T1: control (100% of soil field capacity); T2: stress (50% of soil field capacity); T3: stress + salicylic acid (SA); T4: stress + methionine application (MET); T5: stress + SA + MET. Ψw: water potential (MPa); RWC: relative water content (%); LA: leaf area (cm^2^); FM: fresh mass (g); TSS: total soluble sugars in leaf blades (mg g^−1^ FM); PRO: free proline in leaf blades (µmol g^−1^ FM); TFAAs: total free amino acids in leaf blades (µmol g^−1^ FM); TSPs: total soluble proteins in leaf blades (mg g^−1^ FM); H_2_O_2_: hydrogen peroxide in leaf blades (µmol g^−1^ FM); APX: ascorbate peroxidase activity in leaf blades (nmol of ascorbate min^−1^ mg^−1^ of protein); CAT: catalase activity in leaf blades (µmol of H_2_O_2_ min^−1^ mg^−1^ of protein); SOD: superoxide dismutase activity in leaf blades (U min^−1^ mg^−1^ of protein); A: net photosynthesis (µmol CO_2_ m^−1^ s^−1^); E: transpiration (mmol H_2_O m^−1^ s^−1^); gs: stomatal conductance (mmol H_2_O m^−1^ s^−1^); Ci: internal CO_2_ concentration (ppm); PC: principal component. Genotype means followed by the same uppercase letters do not differ by Student’s *t*-test (*p* > 0.05), and treatment means followed by the same lowercase letters in the rows do not differ by Tukey’s test (*p* > 0.05).

## Data Availability

Not applicable.

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
