# Peer review of "Osmoregulatory and Antioxidants Modulation by Salicylic Acid and Methionine in Cowpea Plants under the Water Restriction"

_plants, 2023, doi:10.3390/plants12061341_

Round 1

Reviewer 1 Report (Previous Reviewer 1)

review-plants-2279731

The authors have improved the manuscript a lot, but a few more sentences need to be checked, these include:

162-168 lines: this part seems to be written by smaller font size than the other parts

197 line: TSP increased in BRS Novaera under stress compared to the control and not decreased. Need to check.

2011 line: „H2O2 increased by 30%    ”  I think it is decreased, seen on table 2

2027 and 229 lines:  Comparison of the percentages values for SA and CAT activity corresponds to the stress conditions not the control. Check it.

238 line: „increased by 45% for BRS Novaera…” This has been reduced as seen in Table 2. Need to check

240 line: Sentence valid after 8 days of treatment? To be communicated

298 line: „reduce..” this is true at 16 day after treatment but not for the 8 days. Check it

Between lines 413-422, the 36 reference occur twice, but there are no 37 reference given in the text. It needs to be checked. 

Typos, they need to be corrected:

24 line: „ Sixteen” to sixteen

164 line: „AL” to LA,

167 line and Table 2 below in the note: „MF” to FM

207 line: „AS” to SA (abbreviation of salicyl acid)

Author Response

Dear Reviewer 1#

We have taken all of your suggestions and have adjusted the text in the lines: 24, 162-168, 196-197, 207, 228, 230, 239-240, and 299.

All changes are underlined in red in the text.

We appreciate your recommending our manuscript for publication.

Best regards,

Alberto Soares de Melo, D.Sc. and Professor

Reviewer 2 Report (Previous Reviewer 2)

The authors significantly improved the manuscript.

Author Response

Caro Avaliador 2#

Agradecemos sua recomendação de nosso manuscrito para publicação.

Atenciosamente,

Alberto Soares de Melo, D.Sc. e Professor

Reviewer 3 Report (Previous Reviewer 3)

The manuscript has been improved especially by adding Table 2.

The problem is what part was used for measuring TSS, Pro, TFAA, H2O2, APX, CAT, and SOD. I didn't find the analytical parts, such as leaf blades, leaf blades+petioles, shoots, or something else. Please write the analytical parts in materials and methods, and Table 2.

Author Response

Dear Reviewer 3#

We have taken all of your suggestions and have adjusted table 2 and the text in the lines: 473, 475, 479, 486, 488, and 491.

All changes are underlined in red in the text.

We appreciate your recommending our manuscript for publication.

Best regards,

Alberto Soares de Melo, D.Sc. and Professor

This manuscript is a resubmission of an earlier submission. The following is a list of the peer review reports and author responses from that submission.

Round 1

Reviewer 1 Report

plants-2151528

General comments

Authors studied the effect of salicylic acid and methionine treatments on osmotic regulation and antioxidant enzyme activity of two cowpea varieties under limited water supply. Figure 1 and 2 are expressive but only a general conclusion can be drawn from them. Figure A and C illustrate the different reactions of the varieties to treatments but Figure B and D show physiological reactions only in general. However, from these figures (1B,D and 2B,D respectively) it is not possible to determine the osmotic and physiological reaction of the two varieties according to PC1-PC4 which the authors claim. In the textual assessment, there is no separation of the data in table and figures. In many spaces, in the discussion section one cannot distinguish one’s own result from that of others. 

Table 2 is too crowded and there is no evaluation of them in the text.

Propose to revision of Table 2:

Only the data of the varieties and the statistical difference between them should be presented and evaluated during treatment with T1-T4. The statistical comparison of the scores of principal components according to treatments should be presented in both cases (after 8 and 16 days) for the two varieties in the supplement Table and not here and refer to near figures 1 and 2. 

Detailed comments

page 2, 91 line: PC1 (36%) is more like 38% according to Table 1

In 93 line, the abbreviation of stomatal conductance is wrong, the correct gs as in Table 1

page 3, 104-105 lines: “ in PC1, water stress decreased Ψw, LA FM…” What does Table 1 or fig. 1 PC1 refer to? It should be checked.

112-113 lines: “..increased RCW, H2O2..” Is it sure? I see these have been reduced as Fig.1 B shows

page 4, 115 line: the statement applies more to Table 1 than to Fig.1 C and D. It needs to be checked

116-125 lines: Fig.1 C and D do not show the claim for G1 and G2 varieties. A reference could be made here to the supplement Table containing statistical analysis of principal components data.

A similar approach should be taken for evaluation of Fig. 2 when the authors present the differences in PCs for G1 and G2 varieties.

page 5, 131 line: gs is not in Fig. 2B and D

in 139 line in the note below Fig. 2 the (CPs) is wrong, PCs are correct?

139 line: “On the other hand, stress imposition in G2…” I think Table 2 should be referenced here rather than Fig 2 C,D.

page 6, 154-156 lines: after editing Table 2, the text about the principal components should be deleted from the title i.e. the scores of the principal components. 

In the discussion section, separate the results obtained for different plant species from own results in the following places:

page 8, 198 line: 22 reference concern on the Populus sp. not on cowpeas. There is no separation here what is own result.

page 8, 203-204 lines: the finding on lipid peroxidation seems to belong to others.

page 8, 212-216 lines: (ref.23) refer to Achillea and not cowpea

in lines 225-227, references should be moved after “under water stress restriction”

on page 9, I propose to redefine the sentences in the following places: lines: 242-244, 266-267 (ref. 11), 272 (ref. 12)

Other comments

page 7, 174 line: Is TSP the correct one instead of TTSP?

in 199 line: SOD occurs 2 times, should be deleted

page 9, 237 line: is 0.21gL-1 SA? Here it is also necessary to specify.

in references, the 37 reference is not in the text.

Reviewer 2 Report

The presented manuscript contains interesting, valuable and original results concerning the possibility of using plant elicitors (salicylic acid and methionine) as the metabolism modulators of the cowpea plants which mitigated the effects of water stress. Taking into account the global climate changes this topic is very important. The authors proved that the application of salicylic acid and methionine could modulate cowpea plants' osmotic and antioxidant metabolism under water restriction.

The objective of this paper is clear and properly defined. The results are clearly presented.

The results are well presented using both charts and tables. The discussion of the results is quite good. The description of the results is correct and the discussion is adequate.

Moreover, the methodology of this paper is appropriate. The used

methods are adequate and the statistical analysis of the results is properly performed.

Conclusions are supported by the obtained results and prove the stated hypothesis. Further research is required to study the proposed elicitors as production indicators of the species, including under field conditions as modulators for different cowpea cultivars.

The listed references are appropriate and properly used.  

To sum up, the manuscript is well-written with understandable and good English and contains important data with practical application. I have only two minor comments.

Line 59, 81, and in the whole manuscript: Please explain the abbreviation before the first time using it.

Line 199 “SOD activity SOD”

Reviewer 3 Report

The purpose of this research is reasonable, but the manuscript needs major revision.

Materials and Methods;

Line 308-310: Please add the details of soil fertilization and amendment. What kind of and how much materials did you use? What was the final pH in the soil after correction? 

Line 315: The authors used a pot experiment but the pot size and the weight or volume of soil have not been described. Please add these. Cowpea is a legume plant with root nodules to fix N2, did you inoculate the rhizobia?

How many pots did you use for one treatment (N=?)? Replication.

Results;

The authors started with the statistical analysis in Table 1 and Figures 1 and 2, then showed the data in Table 2. I would like to recommend that the data should be shown first, then statistics.

LA and FM are the most important indicators for growth, but the SA and MET treatments did not necessarily recover them.

It is strange that LA and FM of the G2-BRS Novaera decreased from 8 DAT to 16 DAT.

Please add the statistical evaluation for each parameter among the 5 treatments by Tukey's method.

Mis typing;

L64: 7.2 M of AS >  7.2 Mg of AS?

L156: 15 days  > 16 days

L174: TTSP > TSP

L199: SOD activity SOD > SOD activity]

Reviewer 4 Report

Review comments:

The research paper “Osmoregulatory and antioxidants modulation by salicylic acid
and methionine in cowpea plants under the water restriction” presents the results of effect of exogenous applications of salicylic acid and methionine on the osmoregulation metabolism

under water restriction in cowpea plants. Free amino acids, proline, soluble proteins, free carbohydrates, and the activity of the enzymes SOD, APX, and catalase were mainly focused. However, the displayed information of the research seemed unsatisfactory, and much important information especially the presentation of experimental data was not properly handled that maybe need major correction.

Abstract:

      The results part was disposed roughly. Thus I suggest the authors to display the results of the study concretely instead of just giving three conclusive sentences.

 Introduction:

      1. There were too much paragrphs in this part and the contents were relatively independent. They were suggested to be rearranged and the fragmented parts should be reintrgrated;

      2. The authors introduced some research results of previous study, yet failed to emphasize the innovation of the current study.

Materials and Methods:

      As to the soil moisture monitoring method, the experiment was designed to irrigate 50% water volum lost by evapotranspiration every day. I am skeptical about the condition that was it really a water deficit status for the cowpea plants in this study?

Results

      1. In line 81-90, these explanations was too redundant and should not be described here; there was no need to displayed both full name and abbreviation in Table1;

       2. There were serious defects in the presentation of experimental data. A number of physiological mesurements  were involved in the study, however no detailed results were displayed. Only some correlation analysis were conducted without foundation.

Discussion

    The same problem as in the “ Introduction” part. Please integrate little paragraphs and make clear viewpoints.